# Factors associated with female age at first marriage: An analysis using all waves of the Pakistan Demographic and Health Survey

**Afza Rasul**[1,2], **Jamal Abdul Nasir**[2]*, **Sohail Akhtar**[3], **Andrew Hinde**[4]

**1** Department of Statistics, Lahore College for Women University, Lahore, Pakistan, **2** Department of Statistics, Government College University, Lahore, Pakistan, **3** Department of Mathematics and Statistics, University of Haripur, Haripur, KPK, Pakistan, **4** Southampton Statistical Sciences Research Institute, University of Southampton, Southampton, United Kingdom

* dr.jamal@gcu.edu.pk

**Data Availability Statement:** The data used in this paper come from the 1990-91, 2006-07, 2012-13, 2017-18 Pakistan Demographic and Health Survey. The files are located at the following urls: https://

## Abstract

In conventional Muslim societies, fertility occurs within the marital union. Therefore, fertility outcomes depend on females' age at first marriage (AFM). The present study explores the pattern of AFM in Pakistan, analyses of which are scarce in the literature. We aim to identify the factors associated with the AFM among currently married women in Pakistan. Demographic factors like birth cohort, and socioeconomic and cultural factors such as province and place of residence, education level, whether or not a woman had ever worked before marriage, ethnicity, and husband's education were studied to explore the pattern of female AFM. Data were taken from the Pakistan Demographic and Health Surveys (PDHSs) of 2012–13 and 2017–18, and a comparison was made with the findings from the earlier PDHSs of 1990–91 and 2006–07. The analysis concentrates on women born between 1941 and 1992, aged between 25 and 49 years during the data collection periods. One-way analysis of variance (ANOVA) was used to assess the difference between the mean AFM for different sub-groups of the population. To identify the covariates that are associated with AFM multiple linear regression models were estimated. We observed a gradually increasing trend in female AFM over time among women born after 1950. The ANOVA results revealed that birth cohort, province, and place of residence, female education level, whether or not a woman had ever worked before marriage, ethnicity, and husband's education were significantly associated with AFM ($p$-values < 0.05). In a multiple regression model, we found that the birth cohort significantly affects the AFM ($p$-value < 0.05). Having worked before marriage is associated with a statistically significant one-year rise in the AFM. Interestingly, all other ethnic groups have lower AFM compared with women whose mother language was Punjabi. Education has a highly significant effect on the AFM: the regression results revealed that uneducated females have a mean AFM 4 to 5 years lower than highly educated women. The results also revealed that educated men marry older women as compared to uneducated men. We conclude that the education of females and even males in Pakistan could lead to a rise in the female AFM.

dhsprogram.com/data/dataset/Pakistan_Standard-DHS_2017.cfm?flag=1 https://dhsprogram.com/data/dataset/Pakistan_Standard-DHS_2012.cfm?flag=0 https://dhsprogram.com/data/dataset/Pakistan_Standard-DHS_2006.cfm?flag=1 https://dhsprogram.com/data/dataset/Pakistan_Standard-DHS_1991.cfm?flag=1.

**Funding:** No financial support is received for this work.

**Competing interests:** It is submitted that there is no competing interests among the authors.

## Introduction

In conventional Muslim societies like Pakistan, fertility occurs after marriage. Marriage marks the onset of the socially acceptable time for childbearing. It is a well-known fact that women who marry early will have, on average, a longer period of exposure to the risk of pregnancy, often leading to higher fertility. Therefore, the female age at first marriage (AFM) is a principal determinant of the number of births a woman will have. Previous work has shown that societies with later AFM have experienced decreased fertility rates [1–3] while, in traditional populations in Asia and Africa, where the AFM is younger, high levels of fertility have been observed [4–8]. Some findings of meta-analysis from 15 developing countries suggested that women marrying before 18 years have a roughly 50 per cent higher chance of contributing to high fertility compared with women married after or at the age of 18 years [9]. Thus, early female marriage will directly influence the fertility rate of the country leading to population growth. Moreover, girls who get married before the age of 18 years have a lower chance to get a higher education, a greater chance of facing unwanted pregnancies, and a higher risk of reproductive health morbidity and maternal mortality [10]. Global statistics show that South Asia has the largest number of child brides as about 285 million girls marry before their 18th birthday [11].

Our objectives in this paper are (1) to chart trends in the mean age at first marriage for women in Pakistan born since about 1940 and marrying since around 1960; and (2) to identify the factors associated with the age at first marriage for women in Pakistan among women born and marrying for the first time during the same periods. The analysis of female AFM remains a neglected dimension in the population and demographic research on Pakistan. The present study will examine the existing situation of female AFM in Pakistan and estimate the association between some socio-demographic and cultural factors and female AFM. The study will use all four waves of the Pakistan Demographic and Health Survey (PDHS): 1990–91, 2006–07, 2012–13 and 2017–18. Different demographic (birth cohort), and socio-demographic and cultural (province of residence, place of residence, education level, whether a woman had ever worked before marriage, ethnicity, and husband's education) characteristics are studied to explore the pattern of female AFM in the country.

Pakistan is an Islamic Republic having four provinces (Punjab, Sindh, Balochistan, and Khyber Pakhtunkhwa (KPK)—the latter being previously known as North West Frontier Province), two autonomous territories (Azad Jammu and Kashmir (AJK) and Gilgit-Baltistan), one federal territory (Islamabad), and one narrow strip territory (Federally Administrated Tribal Areas (FATA)). Punjab is the most populated province of Pakistan having more than 50% of the population. According to the Pakistan Census Report 2017, about 64% of the Pakistani population lives in rural areas. Among the provinces, Sindh is the most urbanized and Baluchistan is the least urbanized province. Urdu is the national language of the country, but all provinces kept their languages according to their cultural and ethnic norms. In Punjab, Punjabi and Siraiki are the most spoken languages. The Sindhi language is the major language of Sindh province and Baluchistani people belong to Balochi and Barauhi ethnic groups. The major language in KPK is Pushto.

Pakistan is the world's fifth most populous country with a population of more than 212.2 million. The female legal age at first marriage in Pakistan was, until 2019, 16 years. In April 2019, the Senate of Pakistan passed a bill stipulating that legal age of first marriage for females will be 18 years. Even though the legal age for marriage was 16 years in Pakistan, many females got married in their childhood i.e. at ages below 16 years [10, 12, 13]. Pakistan is a developing nation with a low literacy rate. According to the Pakistan Economic Survey, 2018–19, the female literacy rate in Pakistan is 55.8% (Punjab 57.4%, Sindh 49.9%, KPK 38.5% and Balochistan 33.5%).

Traditionally, in Pakistan, marriage is a family institution, so that when and to whom a woman gets married is decided by her family. However, in recent decades the decisions of families are being influenced by a range of social, economic and cultural factors, and these can potentially have an association with the age at first marriage. All over the world, much research has focused on the relationship between the timing of female AFM and education level. Researchers found a strong correlation between educational status and females AFM. In Asia, the general conclusion is that females with higher education tend to delay their marriages compared with women with no or low education [13–20] and the same is true of developing countries worldwide [21–25]. Median age at marriage is also found higher in women with secondary education when compared to the uneducated in Malawi [26] Nepal [27] and Myanmar [28]. Additionally, the husband's education level has been a positive influence on the timing of female AFM as it also contributes to the delay in the early marriage of females [17, 19, 20, 22, 24].

Similarly, one more prominent factor which has been associated with a delay in female AFM is women's participation in the labor force. Women who work before marriage tends to delay their marriage [16, 17] due to searching for some suitable match or due to some employment promises. Other variables that have been associated with variation in female AFM are place of residence and ethnicity. Many researchers focused on the place of residence (urban/rural) [16, 19, 20, 25]. They have generally found that women living in rural areas are more likely to marry early that are women from urban areas [16, 20, 25]. The association between urbanization and late AFM is to be expected as women living in urban areas often have more opportunities to get better education and employment than rural residence women. Additionally, previous work has found regional variation in the female AFM in Pakistan. Nasir (2013) found that Sindh has the lowest mean AFM in all four provinces of Pakistan [13]. Similar regional variations have been also observed in other Asian countries [24] and worldwide [21, 23, 29].

Ethnicity has also been found to be an important predictor of age at first marriage in several studies [30]. Ethnic differentials in the timing of marriage have been observed in Pakistan [13], as well as in other parts of the worlds like Bangladesh [24, 31], Malawi [21], Ethiopia [32] and Nepal [27, 33].

## Material and methods

The present research uses the data from nationally representative household surveys, namely Demographic and Health Surveys (DHSs). In Pakistan, four DHSs have been conducted between 1990 and 2018 coordinated by the National Institute of Population Studies (NIPS), Islamabad, and named as Pakistan Demographic and Health Surveys (PDHS) [34–37]. These DHSs are cross-sectional surveys that use several different sets of questionnaires. One of the questionnaires is a women's questionnaire, used to collect information from ever-married women aged 15–49 years at the time of interview. Based on this women's questionnaire survey, analysis of the data from currently married females aged between 25–49 years from two PDHSs (2012–13 and 2017–18) has been undertaken. We explain in the next section why we do not include women aged 15–24 years in our analysis.

The results are compared with earlier findings from Nasir (2013) based on two earlier PDHSs (1990–91 and 2006–07) to analyze trends in the AFM of females from Pakistan. Taken together, our analysis and the earlier analysis of Nasir (2013) include females born in the period 1941 to 1992 (and marrying roughly in the period 1955 to 2016). Table 1 shows the study sample.

The dependent variable in our analysis is female age at first marriage. In the PDHS 2012–13 and PDHS 2017–18, age at first cohabitation is given, which we use as a proxy of age at first

**Table 1. Study Sample in all waves of Pakistan Demographic and Health Survey (PDHS).**

| Survey | Number of Eligible Women | Number of Women Successfully Interviewed | Response Rate (%) | Study Sample Currently, Married women aged 25–49 | Birth Years of the study sample |
|---|---|---|---|---|---|
| PDHS 1990–91 | 6904 | 6611 | 95.7 | 4946 | 1941–65 |
| PDHS 2006–07 | 10601 | 10023 | 64.5 | 7477 | 1957–81 |
| PDHS 2012–13 | 14569 | 13558 | 93.1 | 10441 | 1963–87 |
| PDHS 2017–18 | 15930 | 15068 | 94.6 | 11590 | 1968–92 |

marriage. We examine the distribution of ages at first marriage in two ways: first, using the mean and, second, looking at the cumulative percentages of women married by exact ages 15, 20 and 25 years. The covariates include birth cohort, place of residence, region/province of residence, education of the respondent, whether the respondent had ever worked before marriage, ethnicity (based on mother language), and husband's education. Birth cohort refers to the birth years of females categorized into five-year periods. Place of residence refers to the place of living at the time of the survey, categorized as urban and rural. Region of residence is the province of residence of females (Punjab, Sindh, KPK, Balochistan, Gilgit-Baltistan, Islamabad (ICT), AJK, and FATA). Education of respondents is classified as no education, primary (1–5 years of schooling), secondary (6–10 years of schooling), and higher (11+ years of schooling). Ever worked before marriage categorized as 'yes' (working) and 'no' (not working). Husband's education is classified in the same way as the respondent's education as no education, primary, secondary and higher.

The population of Pakistan has a complex ethnicity but the PDHSs did not ask direct questions about ethnicity. For the surveys of 2006–07 and 2012–13 we have inferred ethnicity from answers to the question 'What is your mother tongue?' The possible answers in 2006–07 were, Urdu, Punjabi, Sindhi, Pushto, Balochi, English, Barauhi, Siraiki, Hindko, Kashmiri, Parahi, Potowari, Marwari, Farsi and other; in 2012–13 additional categories Shina, Brushaski, Wakhi, Chitrali/Khwar and Balti were added. In the present study, we reduced these to eight categories by taking the seven largest linguistic groups as individual categories and combining the remainder into a residual 'others' category.

The present study used IBM SPSS (version 25.0) to manage and analyze the data. Firstly, descriptive analysis for all study variables is performed in the form of means and standard deviation to explore the pattern of AFM according to various study variables. Secondly, to investigate the relationship between the outcome variable (AFM) and independent covariates, bivariate analysis is performed using analysis of variance (ANOVA) and Student's $t$-test statistics. Finally, multiple regression modeling is performed to examine the effect of factors associated with the female AFM.

## Results

Results in this section are based on the analysis of currently married women born in the periods 1941–65 (PDHS 1990–91), 1957–1981 (PDHS 2006–07), 1963–87 (PDHS 2012–13), and 1968–92 (PDHS 2017–18). Table 2 shows the mean age at first marriage for currently married women in the four surveys according to the age of the women at the time of the survey. The mean age at marriage for women in the 2017–18 survey was between 19.4 and 20.0 years for all age cohorts over 25 years. For younger age cohorts the mean age at marriage was substantially lower (18.16 years for those aged 20–24 years and only 16.12 years for those aged 15–19 years).

**Table 2. Mean age at first marriage of currently married women by age at the time of the survey, Pakistan, 1990 to 2018: Four Pakistan Demographic and Health Surveys (PDHSs) (1990–91, 2006–07, 2012–13 and 2017–18).**

| Age cohort | 2017–18 (n = 11590) (25–49) years | 2012–13 (n = 10441) (25–49) years | 2006–07 (n = 7264) (25–49) years | 1990–91 (n = 4832) (25–49) years |
|---|---|---|---|---|
| (45–49) | 19.54 (1419) | 18.78 (1508) | 18.93 (1056) | 19.43 (565) |
| (40–44) | 19.44 (1696) | 18.85 (1674) | 18.67 (1140) | 18.51 (751) |
| (35–39) | 19.93 (2614) | 19.17 (2212) | 18.62 (1513) | 18.47 (960) |
| (30–34) | 19.95 (2774) | 19.58 (2380) | 18.67 (1693) | 17.95 (1150) |
| (25–29) | 19.43 (3077) | 19.27 (2667) | 18.73 (1936) | 17.76 (1406) |
| (20–24) | 18.16 (2187) | 18.07 (2010) | na | na |
| (15–19) | 16.12 (725) | 16.20 (599) | na | na |

These lower ages at marriage for the younger age cohorts are selection effects. Women aged 15–19 years who report an age at marriage in the Demographic and Health Surveys must all have married at ages less than 20 years; those women in the same cohort who are yet to marry will not report an age at marriage. As a consequence, the estimates of the mean age at marriage for this age cohort based on the data in the survey are underestimates of the mean age at marriage that would be reported by the women in the same age cohort were they to be interviewed when those who will eventually marry have all married. The same is true to a lesser extent for women aged 20–24 years. but not for older women, as the vast majority of women in Pakistan will marry before their 25th birthdays. Because of this, we exclude women aged under 25 years from the analysis which follows.

Perhaps the easiest way to assess the trend in the female AFM is to plot each of the mean values in Table 2 against the average birth year of the women concerned. Thus, for example, woman aged 45–49 years in 2017–18 were born between in the years 1967–71, so the average birth year can be regarded as 1969. This plot is shown in Fig 1. Overall, the results from the four PDHSs are consistent and show an increase in the mean age at first marriage for women from just over 18 years in the case of women born around 1960 to close to 20 years for women born around 1990. There is some suggestion of a decline in the age at first marriage among women born before 1960 but this is really based only on one data point.

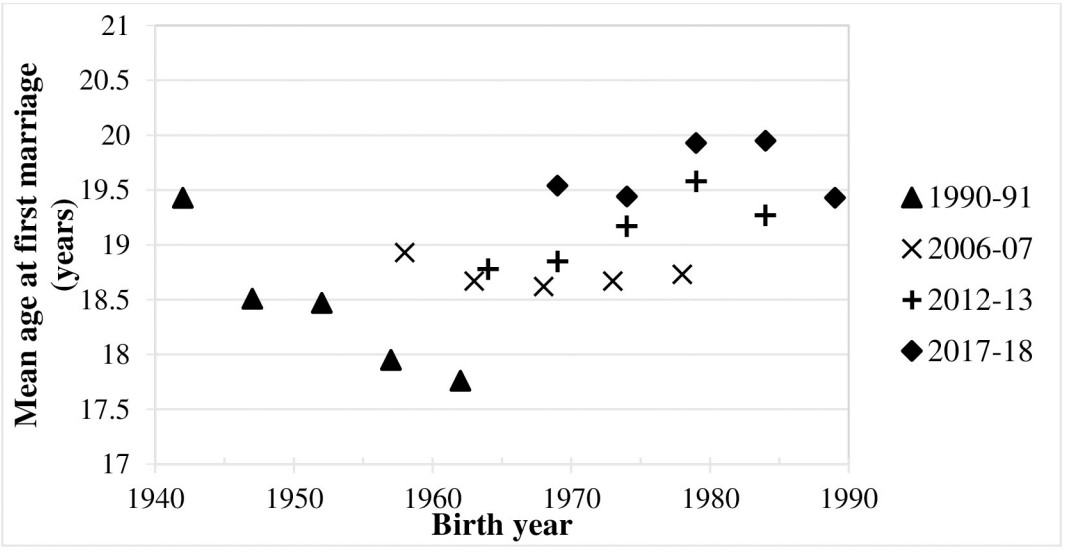

**Fig 1. Mean age at first marriage (AFM)) for women born in different years: Four Pakistan Demographic and Health Surveys (PDHSs) (1990–91, 2006–07, 2012–13 and 2017–18).**

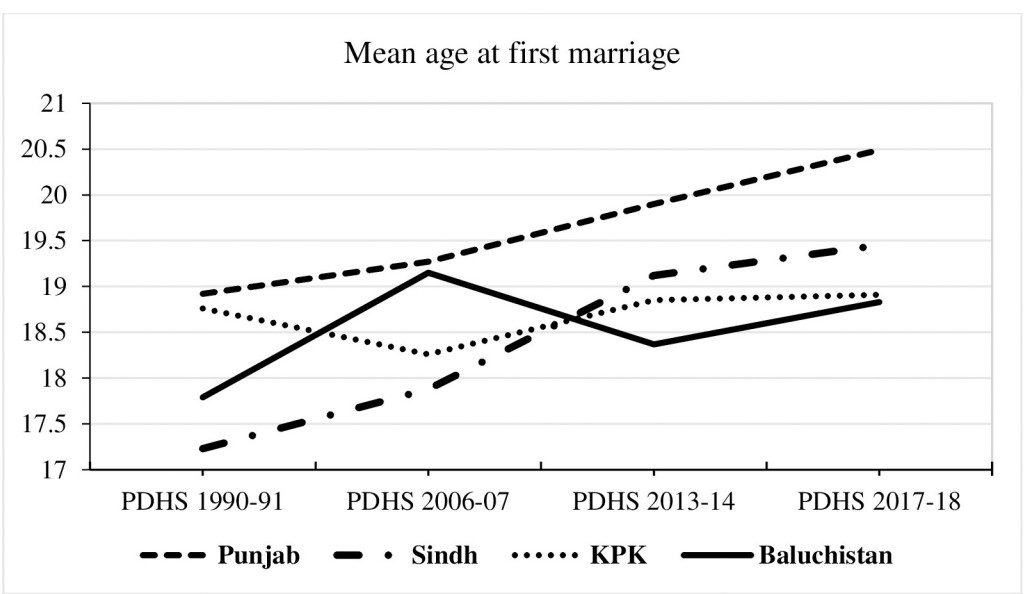

**Fig 2. Mean age at first marriage (AFM)) for currently married women aged 25–49 years in different regions of Pakistan in four Pakistan Demographic and Health Surveys (PDHSs) (1990–91, 2006–07, 2012–13 and 2017–18).**

This result that the AFM is increasing gradually is reinforced by Fig 2 and Table 3, which present the mean AFM for currently married women in the whole country and in each of four regions of Pakistan in the four surveys. In PDHS 1990–91 the female AFM is lowest and in PDHS 2017–18 it is highest.

Table 4 shows the mean AFM by various study characteristics. Females living in urban areas have higher AFM as compared to females belongs to rural areas of Pakistan., Females residing in the province of Punjab are more likely to get married in older age as compared to other provinces. Balochistan shows interesting findings in the PDHS 2006–07, its mean AFM is higher than in all other PDHSs as well as being higher than that in Sindh and KPK in 2006–07. From 2012–13 the PDHS included the separate samples from the female population of Gilgit Baltistan (an autonomous territory) and Islamabad (a Federal territory) and, in 2017–18, separate samples from AJK (an autonomous territory) and FATA (semi-autonomous tribal region) are also included. The descriptive results from all regions of Pakistan showed that females living in Islamabad have the highest AFM (21.22 years, PDHS 2012–13; 21.21 years, PDHS 2017–18) while females belonging to FATA have the lowest AFM (17.96 years, PDHS 2017–18). As FATA is not included separately in PDHS 2012–13, females living in Gilgit Baltistan showed the lowest AFM in all regions of Pakistan for the PDHS 2012–13 (17.34 years).

Education is the most prominent factor associated with the female AFM. Between the PDHS 1990–91 and the PDHS 2017–18 we observed only half-year rise in the female AFM in

**Table 3. Mean age at first marriage (AFM)) for currently married women aged 25–49 years: Four Pakistan Demographic and Health Surveys (PDHSs) (1990–91, 2006–07, 2012–13 and 2017–18).**

| Survey years | Sample size | Age at first marriage (AFM) Mean ± S.D[range] |
| --- | --- | --- |
| 1990–91 | 4946 | 18.24 ± 4.36 |
| 2006–07 | 7477 | 18.68 ± 4.21 |
| 2012–13 | 10441 | 19.18 ± 4.220 [10–48] |
| 2017–18 | 11590 | 19.68 ± 4.367 [10–44] |

**Table 4. Mean age at first marriage of currently married women by various characteristics, Pakistan, 1990 to 2018: Four Pakistan Demographic and Health Surveys (PDHSs) (1990–91, 2006–07, 2012–13 and 2017–18).**

| Characteristics | 2017–18 (n = 11590) (25–49) years | 2012–13 (n = 10441) (25–49) years | 2006–07 (n = 7264) (25–49) years | 1990–91 (n = 4832) (25–49) years |
|---|---|---|---|---|
| **Place of residence** | | | | |
| Urban | 20.20 (5735) | 19.77 (19.77) | 19.11 (2868) | 18.06 (2303) |
| Rural | 19.18 (5855) | 18.63 (18.63) | 18.45 (4396) | 18.44 (2529) |
| **Province of residence** | | | | |
| Punjab | 20.49 (2613) | 19.90 (2906) | 19.27 (3175) | 18.92 (1612) |
| Sindh | 19.45 (2085) | 19.12 (2238) | 17.88 (1868) | 17.23 (1324) |
| KPK[a] | 18.91 (1777) | 18.85 (2027) | 18.26 (1326) | 18.76 (1231) |
| Baluchistan | 18.83 (1312) | 18.37 (1529) | 19.15 (895) | 17.79 (1406) |
| Gilgit Baltistan | 18.67 (769) | 17.34 (943) | - | - |
| Islamabad (ICT[b]) | 21.21 (911) | 21.22 (808) | - | - |
| AJK[c] | 20.74 (1409) | - | - | - |
| FATA[d] | 17.96 (714) | - | - | - |
| **Education** | | | | |
| No | 18.33 (5917) | 18.07 (6021) | 18.08 (4903) | 17.88 (3715) |
| Primary | 19.27 (1566) | 18.84 (1328) | 18.72 (899) | 18.06 (406) |
| Secondary | 20.63 (2303) | 20.31 (1708) | 20.01 (928) | 19.83 (616) |
| Higher | 23.28 (1804) | 22.96 (1384) | 22.31 (534) | 23.69 (95) |
| **Ever worked before marriage** | | | | |
| No | 19.34 (9729) | 18.91 (8290) | 18.64 (5289) | 18.18 (4174) |
| Yes | 21.51 (1858) | 20.23 (2136) | 18.95 (1975) | 18.81 (658) |
| **Ethnicity** | | | | |
| Punjabi | N.A | 20.39 (2421) | 19.68 (2380) | N.A |
| Urdu | - | 21.02 (1059) | 19.94 (577) | - |
| Sindhi | - | 18.73 (961) | 17.55 (866) | - |
| Pushto | - | 18.65 (2230) | 18.69 (1508) | - |
| Balochi | - | 18.00 (426) | 17.84 (390) | - |
| Barauhi | - | 18.01 (456) | 17.78 (112) | - |
| Siraiki | - | 18.41 (962) | 17.43 (911) | - |
| Others | - | 18.41 (1992) | 18.06 (520) | - |
| **Husband Education** | | | | |
| No | 18.29 (3200) | 18.07 (3277) | 17.95 (2740) | 18.03 (2233) |
| Primary | 19.01 (1509) | 18.46 (1346) | 18.24 (1119) | 17.28 (760) |
| Secondary | 19.93 (4033) | 19.34 (3231) | 19.18 (2193) | 18.62 (1462) |
| Higher | 21.28 (2822) | 20.76 (2557) | 20.02 (1212) | 20.20 (377) |

[a] Khyber Pakhtunkhwa.

[b] Independent Capital territory.

[c] Azad Jammu and Kashmir.

[d] Federally Administered Tribal Areas.

uneducated women. Indeed no significant change in the AFM has been found in all four PDHS when we make education constant. But if we analyze keenly, it has been observed that there is a clear change in the proportion of women in the different education categories. A significant rise in the percentage of "higher education" can be observed from earlier (1990–91) to later (2017–18) PDHS descriptive analysis. Similarly, a significant decline can be noted in the

percentage of "no education" from earlier (1990–91) to later (2017–18) PDHS series. Therefore, we conclude that education has played a vital role in delaying the female AFM in Pakistan. Our results showed that the percentage of women working at the time of marriage is low, but they have a half to one-year delay in AFM compared with women who are not working.

Data on ethnicity based on the language spoken was available for PDHS 2006–07 and PDHS 2012–13. Women belonging to Urdu-speaking families have higher AFM followed by women from Punjabi-speaking families. Balochi (PHDS 2012–13, 18.00; PDHS 2006–07, 17.84) and Barauhi (PHDS 2012–13, 18.01; PDHS 2006–07, 17.78) females and Sindhi (PHDS 2012–13, 18.73; PDHS 2006–07, 17.55) and Saraiki (PHDS 2012–13, 18.41; PDHS 2006–07, 17.43) females are very similar in their AFM. Females who married husbands with no education are more likely to marry on average about 2–3 years younger than those who marry husbands with higher education.

Table 5 shows the results of bivariate analysis looking at the association between the covariates and the percentages of women married by exact ages 15, 20 and 25 years for the PDHS 2012–13 and PDHS 2017–18. Test statistic values, probability values ($p$-values) for significant associations, and their degrees of freedom are given below each variable. For the PDHS 2012–13, the birth cohort showed a significant relationship with AFM. In the earliest birth cohort (birth years 1963–67) 23% of females marry before the legal age of marriage in Pakistan; this percentage was lower in the latest birth cohort (birth years 1983–87) but did not vanish. In general, more than 90% of the females marry before the age of 25 years. Place and province of residence, education, whether the woman had worked before marriage, ethnicity, and husband's education are significantly associated with AFM ($p \leq 0.001$). Gilgit-Baltistan showed a remarkable percentage of females getting married before the age of 16 years (36.5%) followed by Baluchistan (22.4%) and KPK (21.2). It is interesting to note that about 24% of the females having no education got married before the minimum/legal AFM while only about 3.5% of the females having higher education got married before the legal AFM in Pakistan. We can see a clear decline in the percentage of women by AFM as the education level increases. The AFM is higher for Urdu- and Punjabi-speaking women compared with other ethnic groups. Results also revealed that about 85% of the Punjabi and Urdu speaking women got married before the age of 25 years while a similar percentage of Balochi and Barahui women got married before the age of 20 years. Males with higher education marry older women. About 24% of uneducated males, married a girl (a female age less than 16 years) while about 89% of educated men married a woman of at least 16 years (legal female AFM in Pakistan) or above. The results for the PDHS 2017–18 are broadly similar, save that the results for the different age groups reveal the clear trend towards later marriage for more recent birth cohorts. For example, in the PDHS 2017–18, only 1.7% of highly educated women got married before the legal AFM while about 9% of males with higher education got to marry a female less than 16 years of age.

To determine the relative importance of covariates that are related to the AFM, two linear regression models are estimated using the two data sets PDHS 2017–18 and PDHS 2012–13. The model for each survey may be written in equation form as follows:

$$Y_i = \beta_0 + \beta_1 A_{i,25-29} + \beta_2 A_{i,30-34} + \beta_3 A_{i,35-39} + \beta_4 A_{i,40-44} + \gamma_1 X_{1i} + \ldots + \gamma_k X_{ki} + \varepsilon_i$$

where $Y_i$ is the age at first marriage of woman $i$; $A_{i,25-29}$, $A_{i,30-34}$, $A_{i,35-39}$ and $A_{i,40-44}$ are dummy variables taking the value 1 if woman $i$ was aged 25–29, 30–34, 35–39 and 40–44 years respectively and 0 otherwise; $X_{1i} \ldots X_{ki}$ are the values of other covariates for woman $i$; and $\varepsilon_i$ is a random, normally distributed error term with a mean of zero. The set of covariates $X_{1i} \ldots X_{ki}$ differed slightly from survey to survey because of data availability. The results from these models have been compared with those in Nasir (2013) [13].

**Table 5. Percentage distribution of currently married women aged 25–49 by various characteristics with age at first marriage [PDHS 2012–13 and PDHS 2017–18].**

| Characteristic | PDHS 2017–18 | | | | PDHS 2012–13 | | | |
|---|---|---|---|---|---|---|---|---|
| | No. of Women | Cumulative percentage married by exact age | | | No. of Women | Cumulative percentage married by exact age | | |
| | | ≤15 | 20 | 25 | | ≤15 | 20 | 25 |
| **Age cohort [a, b]** | | | | | | | | |
| (45–49) | 1429 | 20.8 | 65.1 | 89.7 | 1508 | 23.0 | 75.1 | 92.4 |
| (40–44) | 1696 | 18.9 | 67.0 | 89.2 | 1674 | 18.8 | 75.8 | 93.1 |
| (35–39) | 2614 | 18.7 | 61.2 | 86.0 | 2212 | 18.8 | 70.8 | 90.0 |
| (30–34) | 2774 | 16.3 | 59.0 | 81.2 | 2380 | 16.9 | 65.3 | 88.6 |
| (25–29) | 3077 | 14.7 | 62.5 | 94.9 | 2667 | 15.6 | 64.6 | 96.5 |
| | *F*-Statistic = 9.072; *p*-value = 0.000; *d.f* = 11588 | | | | *F*-Statistic = 11.473; *p*-value = 0.000; *d.f* = 10440 | | | |
| **Place of residence** | | | | | | | | |
| Urban | 5735 | 14.6 | 57.5 | 87.5 | 5054 | 16.0 | 62.9 | 89.2 |
| Rural | 5855 | 20.1 | 67.2 | 92.2 | 5387 | 20.1 | 75.1 | 94.6 |
| | *t*-Statistic = 12.631; *p*-value = 0.000; *d.f* = 11589 | | | | *t*-statistic = 13.936; *p*-value = 0.000; *d.f* = 10439 | | | |
| **Province of residence** | | | | | | | | |
| Punjab | 2613 | 10.7 | 54.1 | 88.2 | 2906 | 13.8 | 62.5 | 89.9 |
| Sindh | 2085 | 19.7 | 63.8 | 90.7 | 2238 | 13.4 | 70.9 | 92.3 |
| KPK | 1777 | 20.7 | 71.6 | 92.7 | 2027 | 21.2 | 72.1 | 93.3 |
| Baluchistan | 1312 | 25.7 | 70.1 | 91.2 | 1519 | 22.4 | 78.8 | 95.8 |
| Gilgit Baltistan | 769 | 25.9 | 70.0 | 93.4 | 943 | 36.5 | 84.2 | 96.4 |
| Islamabad (ICT) | 911 | 9.8 | 46.9 | 82.8 | 808 | 10.0 | 46.0 | 82.8 |
| AJK | 1409 | 9.8 | 53.4 | 85.5 | - | - | - | - |
| FATA | 714 | 26.8 | 80.6 | 96.8 | - | - | - | - |
| | *F*-Statistic = 81.797; *p*-value = 0.000; *d.f* = 11589 | | | | *F*-Statistic = 109.55; *p*-value = 0.000; *d.f* = 10440 | | | |
| **Education** | | | | | | | | |
| No | 5917 | 26.0 | 76.3 | 94.5 | 6021 | 23.6 | 81.5 | 96.2 |
| Primary | 1566 | 15.5 | 67.2 | 93.3 | 1328 | 16.5 | 73.2 | 94.5 |
| Secondary | 2303 | 8.6 | 52.5 | 88.8 | 1708 | 12.0 | 55.0 | 89.1 |
| Higher | 1804 | 1.7 | 24.9 | 72.7 | 1384 | 3.5 | 29.3 | 74.4 |
| | *F*-Statistic = 766.08; *p*-value = 0.000; *d.f* = 11589 | | | | *F*-Statistic = 658.92; *p*-value = 0.000; *d.f* = 10440 | | | |
| **Ever worked before marriage** | | | | | | | | |
| No | 9729 | | | | 8290 | 19.4 | 71.5 | 93.7 |
| Yes | 1858 | | | | 2136 | 13.4 | 60.5 | 85.5 |
| | *t*-Statistic = -19.98; *p*-value = 0.000; *d.f* = 11585 | | | | *t*-statistic = -12.98; *p*-value = 0.000; *d.f* = 10424 | | | |
| **Ethnicity** | | | | | | | | |
| Punjabi | - | - | - | - | 2421 | 11.8 | 53.3 | 84.6 |
| Urdu | - | - | - | - | 1059 | 8.2 | 48.0 | 84.8 |
| Sindhi | - | - | - | - | 961 | 13.8 | 77.7 | 94.1 |
| Pushto | - | - | - | - | 2230 | 22.5 | 73.1 | 93.9 |
| Balochi | - | - | - | - | 426 | 24.9 | 84.3 | 96.0 |
| Barauhi | - | - | - | - | 456 | 22.8 | 86.4 | 97.1 |
| Siraiki | - | - | - | - | 962 | 17.8 | 78.5 | 94.8 |
| Others | - | - | - | - | 1922 | 26.3 | 75.4 | 93.6 |
| | | | | | *F*-Statistic = 92.682; *p*-value = 0.000; *d.f* = 10436 | | | |
| **Husband Education** | | | | | | | | |
| No | 3200 | 27.2 | 76.2 | 94.4 | 3277 | 23.5 | 82.0 | 96.0 |
| Primary | 1509 | 18.4 | 69.2 | 93.5 | 1346 | 20.4 | 76.7 | 94.8 |

*(Continued)*

**Table 5.** (Continued)

| Characteristic | PDHS 2017–18 | | | | PDHS 2012–13 | | | |
|---|---|---|---|---|---|---|---|---|
| | No. of Women | Cumulative percentage married by exact age | | | No. of Women | Cumulative percentage married by exact age | | |
| | | ≤15 | 20 | 25 | | ≤15 | 20 | 25 |
| Secondary | 4033 | 14.8 | 60.1 | 89.5 | 3231 | 17.1 | 66.8 | 92.0 |
| Higher | 2822 | 9.2 | 46.2 | 83.1 | 2557 | 11.4 | 52.0 | 85.3 |
| | *F*-Statistic = 268.84; *p*-value = 0.000; *d.f* = 11563 | | | | *F*-Statistic = 223.39; *p*-value = 0.000; *d.f* = 10410 | | | |

[a] PHDS 2017–18: Birth Cohort (Age Cohort); 1968–72 (45–49), 1973–77 (40–44), 1978–82 (35–39), 1983–87 (30–34), 1988–92 (25–29).

[b] PDHS 2012–13: Birth Cohort (Age Cohort); 1963–67 (45–49), 1968–72 (40–44), 1973–77 (35–39), 1978–82 (30–34), 1983–87 (25–29).

The first two columns of Table 6 display the two final estimated models for currently married women aged 25–49 years old for PDHS 2017–18 and PDHS 2012–13 respectively, while the third and fourth columns display the comparative results from PDHS 2006–07 and PDHS 1990–91. The results for 2012–13 and 2017–18 show that whether a woman lives in an urban or rural area has no significant effect on the AFM. Only the PDHS 2006–07 model showed that females living in urban areas have 0.37-year lower mean AFM as compared to females living in rural areas of Pakistan. In PDHS 2012–13 and PDHS 2017–18, results showed that female AFM is lower in the rural area of Pakistan as compared to urban areas (Table 4) but these results were not statistically significant after controlling for other covariates (Table 6). In the regression models, we took the province Punjab (a highly populated and economically developed province of Pakistan) as a reference category. The results demonstrated that Sindh has a mean female AFM about 0.5–1.5 years lower than that of Punjab in all four PDHSs. Baluchistan showed interesting findings: in the PDHS 1990–91, its mean female AFM was 0.70 years lower than that in Punjab, in PDHS 2006–07 it was amongst one year higher; in the PDHS 2012–13 it was almost the same, while in PDHS 2017–18 it was again about half a year less than that in Punjab. PDHS 2012–13 and PDHS 2017–18 also include a separate sample for Gilgit-Baltistan (on 29 August 2009, the government of Pakistan announced Gilgit Baltistan as a provincial autonomous region). The results showed that Gilgit-Baltistan has a mean female AFM more than 1.5 years lower than Punjab in the most recent two PDHS data sets. Islamabad and AJK show a mean female AFM which is not significantly different from that in Punjab.

The regression models highlighted female education, the most prominent factor associated with the female AFM. All four regression models revealed that, compared with women with no education, women with only primary education have a mean AFM about half a year higher. Secondary education raises the mean AFM by 1.5–2.0 years, and higher education by 4–5 years compared with no education. Working before marriage is also associated with a significant delay to first marriage. Women, who participate in the labor force delay their marriage about half to one year (PDHS 2017–18, 0.94 years; PDHS 2012–13, 0.83 years; PDHS 2006–07, 0.57 years) when compared to non-working women (PDHS 1990–91 showed an insignificant result). All other ethnic groups showed significantly lower female AFM when compared with Punjabi women (Table 6). There is a positive association between the education of the husband and the mean AFM for the women.

## Discussion

The present article focuses on examining the trends and finding the factors affecting AFM of Pakistani currently married females aged between 25–49 years old born in the period 1941–92. Like other Asian countries; Iran [22], Bangladesh, Nepal, India [10], we observed a gradually

**Table 6. Multiple regression estimates of the factors associated with age at first marriage of currently married women (PDHS 1990–91, PDHS 2006–07, PDHS 2013–14 and PDHS 2017–18).**

| Characteristics | PDHS 2017–18 Coefficients (SE) | PDHS 2012–13 Coefficients (SE) | PDHS 2006–07 [a] Coefficients (SE) | PDHS 1990–91 [a] Coefficients (SE) |
|---|---|---|---|---|
| Intercept | **18.909** (0.1410)** | **18.799** (.1493)** | **19.11** (0.16)** | **19.66** (0.20)** |
| **Age Cohort** | | | | |
| (45–49) | RC | RC | RC | RC |
| (40–44) | -0.344* (0.141) | -0.13 (0.135) | -0.36* (0.17) | -0.92** (0.23) |
| (35–39) | -0.003 (0.130) | 0.074 (0.127) | -0.42** (0.16) | -1.03** (0.22) |
| (30–34) | -0.259* (0.129) | 0.264* (0.126) | -0.55** (0.16) | -1.55** (0.21) |
| (25–29) | -0.743** (0.128) | -0.050 (0.125) | -0.60** (0.15) | -1.81** (0.21) |
| **Place of residence** | | | | |
| Rural | RC | RC | RC | RC |
| Urban | -0.091 (.0795) | -0.032 (0.0844) | -0.37** (0.11) | Not significant |
| **Province of residence** | | | | |
| Punjab | RC | RC | RC | RC |
| Sindh | -0.781** (0.118) | -0.605** (0.144) | -0.42** (0.17) | -1.53** (0.15) |
| KPK | -0.950** (0.125) | 0.127 (0.177) | -0.57* (0.22) | 0.05 (0.16) |
| Baluchistan | -0.520** (0.141) | 0.055 (0.185) | 0.98** (0.22) | -0.70** (0.19) |
| Gilgit Baltistan | -1.540** (0.163) | -1.638** (0.206) | - | - |
| Islamabad (ICT) | 0.027 (0.154) | -0.040 (0.163) | - | - |
| AJK | -0.048 (0.131) | - | - | - |
| FATA | -1.153** (0.173) | - | - | - |
| **Education** | | | | |
| No | RC | RC | RC | RC |
| Primary | 0.537** (0.119) | 0.398** (0.122) | 0.41** (0.15) | 0.37* (0.23) |
| Secondary | 1.865** (0.111) | 1.741** (0.122) | 1.58** (0.16) | 1.90** (0.21) |
| Higher | 4.205** (0.136) | 4.108** (0.151) | 3.87** (0.22) | 5.38** (0.48) |
| **Ever worked before marriage** | | | | |
| No | RC | RC | RC | RC |
| Yes | 0.939** (0.106) | 0.832** (0.099) | 0.57** (0.11) | Not significant |
| **Ethnicity** | N.A | | | N.A |
| Punjabi | - | RC | RC | - |
| Urdu | - | -0.163 (0.164) | -0.40* (0.21) | - |
| Sindhi | - | -0.709** (0.193) | -1.50** (0.22) | - |
| Pushto | - | -1.035** (0.183) | -0.32 (0.23) | - |
| Balochi | - | -1.557** (0.243) | -1.83** (0.28) | - |
| Barauhi | - | -1.237** (0.256) | -1.99** (0.42) | - |
| Siraiki | - | -1.229** (0.156) | -1.79** (0.17) | - |
| Others | - | -0.829** (0.170) | -1.42** (0.27) | - |
| **Husband Education** | | | | |
| No | RC | RC | RC | - |
| Primary | 0.256* (0.126) | 0.034 (0.125) | 0.22 (0.14) | -0.58** (0.18) |
| Secondary | 0.587** (0.102) | 0.371** (0.102) | 0.56** (0.12) | -0.08 (0.15) |
| Higher | 0.680** (0.122) | 0.489** (0.123) | 0.52** (0.16) | 0.39 (0.27) |
| **Regression Model fitting** | | | | |

(*Continued*)

**Table 6.** (Continued)

| Characteristics | PDHS 2017–18 Coefficients (SE) | PDHS 2012–13 Coefficients (SE) | PDHS 2006–07 [*a] Coefficients (SE) | PDHS 1990–91 [*a] Coefficients (SE) |
|---|---|---|---|---|
| | $F$-Statistics = 145.53 | $F$-Statistics = 105.32 | - | - |
| | $d.f$ = 19 | $d.f$ = 24 | | |
| | $p$-value = 0.000 | $p$-value = 0.000 | | |
| | $R^2$ = 0.193 | R-Square = 0.196 | | |
| | Adjusted $R^2$ = 0.192 | Adjusted $R^2$ = 0.194 | | |

[**] P-value <0.01

[*]0.05> P-value ≥0.01 RC: reference category

[*a] Nasir (2013).

increasing trend in female AFM with the passage of time in Pakistan. Though the mean AFM increases just 1.44 years over time (from PDHS 1990–91, mean AFM = 18.24; to PDHS 2017–18, MAFM = 19.68). This increase was observed to be significant after controlling for socioeconomic and cultural factors. As the education level of females increases, AFM also increases. Our result revealed that women with primary education tended to marry about one year later than uneducated women. The study findings suggest that education is a highly significant factor in delaying AFM in Pakistan as in other Asian [15, 17, 19, 20, 22, 27, 33] and worldwide [21, 23–25, 29] studies. Results of the analysis showed that college or university education was associated with a mean female AFM than four years greater than that for women with no education. Though this relationship between higher education and delayed marriage is a little complex, continued schooling/college of females results in the delay at the timing of AFM. This is a very informative finding for Pakistani society, where mean female AFM is quite young—less than 18 years in many areas. The results also revealed that the husband's education may plays a significant role in the delay of the AFM of females. This may be since males with higher education are more likely to marry mature women rather than a girl or they want to marry a more educated woman. Similar findings were observed in other Asian countries [17, 19, 22]. Thus, the longer the boys and girls, stay in school/education, the less likely they are to be married at an early age. Women's participation in the labor force before marriage is an important determinant that significantly affects the AFM in Pakistan and other regions of the world [19]. Results showed that a female getting higher education, involved in the labor force before marriage and wants to marry a highly educated man got married more than 6 years later than a woman with no education, who did not work before marriage and who married an uneducated man.

In many Asian researches, female AFM is lower in rural areas of the country [16, 20, 24]. The present study did not show any significant effect of place of residence at AFM except in PDHS 2006–07, wherein rural areas mean AFM is higher as compared to urban areas. A similar result is also observed in India [19]. Region of residence significantly affect the mean AFM in Pakistan and these regional variations are also observed in Bangladesh [24], being ascribed to cultural norms and family preferences.

Ethnicity has a large impact on the timing of marriage in Pakistan. In the present study, the population was divided on the basis of language into eight ethnic groups according to cultural norms. The findings of research concluded that AFM was, on average, about 1.0–1.5 years lower in other ethnic groups when compared to "Punjabi" women. A possible explanation is that in Pakistan, in some ethnic groups it is still held that a female should marry at the earliest age. Ethnicity is also found to be a significant social determinant that has a strong impact on

female AFM in other South Asian and neighboring countries of Pakistan, such as Bangladesh [31], India [19], Nepal [27].

## Conclusion

In conclusion, the present study has shown that, despite the restriction on the female legal age at first marriage, the AFM in Pakistan is relatively low. The study showed that the female AFM has risen slowly among women born since around 1960. The findings suggest that three socio-economic variables (education, female work before marriage, and husband's education) have important associations with AFM. Since fertility is affected by AFM, these variables will have an association with population growth. If the Government of Pakistan wishes to restrict population growth, the following policies will aid that objective. Firstly, confine the legal age at which a girl can marry as 18 years and implement this law vigorously. Secondly, increase the level of female education to ensure at least secondary education for both males and females to get a significant impact on female AFM. One implication of this study is that encouraging a longer period for education or skill-based education among females of Pakistan may well reduce the proportion of early female marriages. Hence, and third, Government and non-Government organizations should look to create skill-based education and job opportunities for females. Motivating females to get involved in skill-based education and, then, in the labor force is one of the key challenges for policymakers. As Pakistan is a patriarchal society, literate males can play an important role in the society to reduce early female marriages. It may be a more successful approach to educate the male of the patriarchal nature of Pakistani society to increase the female age at first marriage. Finally, to reduce the rigid cultural norms/customs in many ethnic groups of Pakistani society about early female marriages, public awareness through the media of Pakistan might be helpful to decide the proper timing of the female age at first marriage.

## Author Contributions

**Conceptualization:** Jamal Abdul Nasir.

**Data curation:** Afza Rasul, Sohail Akhtar.

**Investigation:** Afza Rasul.

**Methodology:** Afza Rasul, Jamal Abdul Nasir, Sohail Akhtar.

**Software:** Afza Rasul, Sohail Akhtar.

**Supervision:** Jamal Abdul Nasir.

**Writing – original draft:** Jamal Abdul Nasir.

**Writing – review & editing:** Jamal Abdul Nasir, Andrew Hinde.

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
