## [Decision Letter · Decision Letter 0]

12 May 2021

PONE-D-21-08432

Factors associated with female age at first marriage using all waves of Pakistan Demographic and Health Survey

PLOS ONE

Dear Dr. Nasir,

Thank you for submitting your manuscript to PLOS ONE. After careful consideration, we feel that it has merit but does not fully meet PLOS ONE’s publication criteria as it currently stands. Therefore, we invite you to submit a revised version of the manuscript that addresses the points raised during the review process.

This manuscript need major revisions.

We look forward to receiving your revised manuscript.

Kind regards,

Faisal Abbas, PhD

Academic Editor

PLOS ONE

Journal Requirements:

2) Thank you for stating the following in the Acknowledgments Section of your manuscript:

[The author is grateful to the Higher Education Commission of Pakistan (HEC) for awarding a PHD scholarship under the Faculty Development Program (FDP) for Pakistani Universities under award Pin No.119-FPS2-005, Dated: 11-12-2019.]

 [No financial support is received for this work]

3) We suggest you thoroughly copyedit your manuscript for language usage, spelling, and grammar. If you do not know anyone who can help you do this, you may wish to consider employing a professional scientific editing service.  

Additional Editor Comments (if provided):

Major revisions.

Reviewers' comments:

Reviewer's Responses to Questions

**Comments to the Author**

1. Is the manuscript technically sound, and do the data support the conclusions?

Reviewer #1: Partly

Reviewer #2: Yes

2. Has the statistical analysis been performed appropriately and rigorously? 

Reviewer #1: Yes

Reviewer #2: Yes

3. Have the authors made all data underlying the findings in their manuscript fully available?

Reviewer #1: Yes

Reviewer #2: No

4. Is the manuscript presented in an intelligible fashion and written in standard English?

Reviewer #1: Yes

Reviewer #2: Yes

5. Review Comments to the Author

Reviewer #1: Line 70-71: Add % sign with all provincial estimates

Line 156-159: “The reason behind this is interpretable, as, in later birth Cohorts, the current age is lower, so the females from these lower age groups have obviously lower mean AFM as compared to the females from earlier birth cohorts with the higher current age group.”

Age at first marriage doesn’t change with the increase in age of a particular birth cohort. The lower age groups may have lower AFM but it has no linkage with the current age. If a woman is first married at the age say 20, her AFM will remain at a constant value of 20 regardless of her current age at any point in time.

Line 159-163: “This could be the reason behind the scenario that we have not included current (age at the time of interview) age groups 15-19 and 20-24 in our analysis. As we are studying AFM, and our sample is only married women so if we include females having age groups at the time of interviews 15-19 and 20-24, then we cannot generalize our results to the whole of the female population.”

The sample of the study is currently married women at the time of survey and not the currently married women who were at their legal age at the time of first marriage. Even this case justifies exclusion of only under 18 category. A large fraction of females are married around their mid- twenties in Pakistan. So all females currently married at the time of survey should be included in the study to ensure generalization of results.

It is recommended to include the female belonging to age groups 15-24 in the analysis and update all tables and discussion accordingly.

Reviewer #2: The main contribution of the study is that it studies the pattern of age at firm marriage (AFM) in Pakistan. It attempts to identify different demographic, social and cultural factors and examine their impact of AFM. This is an interesting topic however, following observations are needed to be addressed before acceptance.

1. The author should explicitly identify the objectives of the study. In addition, what methods have been adopted to meet the objectives?

2. In introduction part, although, the author gave references about the studies which examine the relationship between AFN and other variable. However, the study does try to establish any theoretical relationship. It would be better if some theoretical connections are established in addressing the relationship.

3. On page 7, line 97-99, the author mentioned that there are more than twenty ethnic groups in Pakistan and the researcher divided these groups into eight categories. The author did not specify the criteria how these categories have been merged.

There are more

4. In table 5, the author showed the results of multiple regression model however, no regression equation has been presented in material and method part. It would be appropriate if the author develop a regression equation which incorporates both dependent and independent variables. Moreover, the value of R-square and F-Statistic should also be mentioned in the table -5.

6. PLOS authors have the option to publish the peer review history of their article (what does this mean?). If published, this will include your full peer review and any attached files.

Reviewer #1: No

Reviewer #2: No

---

## [Author Response · Author response to Decision Letter 0]

16 Jul 2021

Factors associated with female age at first marriage: an analysis using all waves of Pakistan Demographic and Health Survey

Response to Reviewers

We thank the reviewers of the original version for their constructive and detailed comments. We list these below, point by point, with our responses.

Reviewer #1

 Line 70-71: Add % sign with all provincial estimates

We added the suggestion, (ll 75-76 in revised manuscript)

Line 156-159: “The reason behind this is interpretable, as, in later birth Cohorts, the current age is lower, so the females from these lower age groups have obviously lower mean AFM as compared to the females from earlier birth cohorts with the higher current age group.”

Age at first marriage doesn’t change with the increase in age of a particular birth cohort. The lower age groups may have lower AFM but it has no linkage with the current age. If a woman is first married at the age say 20, her AFM will remain at a constant value of 20 regardless of her current age at any point in time.

Line 159-163: “This could be the reason behind the scenario that we have not included current (age at the time of interview) age groups 15-19 and 20-24 in our analysis. As we are studying AFM, and our sample is only married women so if we include females having age groups at the time of interviews 15-19 and 20-24, then we cannot generalize our results to the whole of the female population.”

The sample of the study is currently married women at the time of survey and not the currently married women who were at their legal age at the time of first marriage. Even this case justifies exclusion of only under 18 category. A large fraction of females are married around their mid- twenties in Pakistan. So all females currently married at the time of survey should be included in the study to ensure generalization of results.

It is recommended to include the female belonging to age groups 15-24 in the analysis and update all tables and discussion accordingly.

We disagree with the reviewer’s suggestion. We are interested in measuring the secular change in the age at first marriage across birth cohorts. However, in the Pakistan Demographic and Health Surveys (DHSs) only ever-married women aged 15-19 and 20-24 years old were interviewed. If, say, we include the women aged 15-19 years in our analysis, then all of them will report ages at marriage below 20 years, and their average age at marriage will be well below 20 years. The women in the same birth cohort—that is, who would have been aged 15-19 years at the date of the survey but who will marry at ages above 20 years will not be included in the sample because they were not interviewed. Because of this truncation of the distribution, our estimates of the mean age at first marriage for women aged 15-19 years at the time of each survey will be biased downwards, possibly severely. This can be illustrated by considering the women who were interviewed in the DHSs who were aged 15 years. All of these were married, else they would not have been interviewed, so they must all have married for the first time at age 15 years or less. 

The same effect will be seen among women aged 20-24 years, though not so severely. However, since almost all women in Pakistan who will ever marry have done so by the time, they are aged 25 years, the bias should be negligible at ages 25 years and over. Hence, we only use data for women aged 25-29, 30-34, 35-39, 40-44, and 45-49 years at each survey in our calculations of the secular trend in the age at first marriage and in our analysis of the factors associated with the mean age at first marriage.

We have added a paragraph in the revised version of the paper to explain this point. (ll 117-127 in revised manuscript)

Reviewer #2 

The main contribution of the study is that it studies the pattern of age at firm marriage (AFM) in Pakistan. It attempts to identify different demographic, social and cultural factors and examine their impact of AFM. This is an interesting topic however, following observations are needed to be addressed before acceptance.

1. The author should explicitly identify the objectives of the study. 

We have added a sentence explicitly to identify these objectives (ll. 45-48 of the revised paper).

In addition, what methods have been adopted to meet the objectives?

The methods are described in ll. 107-64. We have made changes to the wording to make the description of the methods clearer.

2. In introduction part, although, the author gave references about the studies which examine the relationship between AFM and other variables. However, the study does not try to establish any theoretical relationship. It would be better if some theoretical connections are established in addressing the relationship.

We agree with the reviewer that a bit more is needed about the kinds of trends we might expect, and the covariates that we might expect to be associated with the age at first marriage, and we have added this (ll. 77-106). We do not think that a lengthy theoretical section on the determinants of age at first marriage in Pakistan is appropriate for this paper, as our intention is empirically to analyse trends in the mean age at first marriage and in the factors associated with age at first marriage in the country with a view to understanding the likely impact of changes in the age at first marriage on fertility trends.

3. On page 7, line 97-99, the author mentioned that there are more than twenty ethnic groups in Pakistan and the researcher divided these groups into eight categories. The author did not specify the criteria how these categories have been merged.

In the Pakistan Demographic and Health Surveys, direct questions on ethnicity were not asked. In 1990-91 there is no variable in the data from which ethnicity can be easily inferred. For the surveys of 2006-07 and 2012-13 we have inferred ethnicity from answers to the question ‘What is your mother tongue?’ The possible answers were in 2006-07, Urdu, Punjabi, Sindhi, Pushto, Balochi, English, Barauhi, Siraiki, Hindko, Kashmiri, Parahi, Potowari, Marwari, Farsi and other; in 2012-13 additional categories Shina, Brushaski, Wakhi, Chitrali/Khwar and Balti were added. In the present study, we reduced these to eight categories by taking the seven largest linguistic groups as individual categories and combining the remainder into a residual ‘others’ category. We added (ll. 149-56)

4. In table 5, the author showed the results of multiple regression model however, no regression equation has been presented in material and method part. It would be appropriate if the author develop a regression equation which incorporates both dependent and independent variables.

We thank you for this suggestion. We have added a regression equation (ll. 260-69).

Moreover, the value of R-square and F-Statistic should also be mentioned in the table 5.

We thank you for this suggestion. We have added the value of R-square and F-Statistic for regression in the table 5.

Finally, we asked a native English speaker who is also the editor of an academic journal in the English language to copy-edit the paper. The result has been a thorough revision of the language in the text.

---

## [Decision Letter · Decision Letter 1]

16 Dec 2021

PONE-D-21-08432R1Factors associated with female age at first marriage: an analysis using all waves of Pakistan Demographic and Health SurveyPLOS ONE

Dear Dr. Nasir,

Thank you for submitting your manuscript to PLOS ONE. After careful consideration, we feel that it has merit but does not fully meet PLOS ONE’s publication criteria as it currently stands. Therefore, we invite you to submit a revised version of the manuscript that addresses the points raised during the review process.

Major Revisions Please submit your revised manuscript by Jan 30 2022 11:59PM. If you will need more time than this to complete your revisions, please reply to this message or contact the journal office at plosone@plos.org. Please include the following items when submitting your revised manuscript:A rebuttal letter that responds to each point raised by the academic editor and reviewer(s). You should upload this letter as a separate file labeled 'Response to Reviewers'.A marked-up copy of your manuscript that highlights changes made to the original version. You should upload this as a separate file labeled 'Revised Manuscript with Track Changes'.An unmarked version of your revised paper without tracked changes. You should upload this as a separate file labeled 'Manuscript'.

We look forward to receiving your revised manuscript.

Kind regards,

Faisal Abbas, PhD

Academic Editor

PLOS ONE

Additional Editor Comments (if provided):

Major Revisions required.

Reviewers' comments:

Reviewer's Responses to Questions

**Comments to the Author**

1. If the authors have adequately addressed your comments raised in a previous round of review and you feel that this manuscript is now acceptable for publication, you may indicate that here to bypass the “Comments to the Author” section, enter your conflict of interest statement in the “Confidential to Editor” section, and submit your "Accept" recommendation.

Reviewer #1: All comments have been addressed

Reviewer #2: All comments have been addressed

2. Is the manuscript technically sound, and do the data support the conclusions?

Reviewer #1: Partly

Reviewer #2: Yes

3. Has the statistical analysis been performed appropriately and rigorously? 

Reviewer #1: No

Reviewer #2: Yes

4. Have the authors made all data underlying the findings in their manuscript fully available?

Reviewer #1: Yes

Reviewer #2: (No Response)

5. Is the manuscript presented in an intelligible fashion and written in standard English?

Reviewer #1: Yes

Reviewer #2: Yes

6. Review Comments to the Author

Reviewer #1: This study is analyzing trend in age at first marriage in Pakistan across time. The justification provided for missing the age bracket 15-24 is that PDHS only interviews ever married women and “ If, say, we include the women aged 15-19 years in our analysis, then all of them will report ages at marriage below 20 years, and their average age at marriage will be well below 20 years. The women in the same birth cohort—that is, who would have been aged 15-19 years at the date of the survey but who will marry at ages above 20 years will not be included in the sample because they were not interviewed.” Line 117-122.

This argument is not convincing. The generalization of results in any survey based study is restricted to the particular sample selected for the survey. Since the analysis is over time, the claimed inclusion of females who decide to marry above say age 20 is adequately presented in the subsequent survey. Even if this argument is accepted for the age bracket 15-19, it is totally unacceptable for age 20-24. The study itself claims in Line 237 that“In general, more than 90% of the females marry before the age of 25 years”, so inclusion of this age bracket is very less likely to create any downward bias in the estimates.

Reviewer #2: The author has sufficiently addressed my observations. Therefore, from my side, The revision is accepted.

7. PLOS authors have the option to publish the peer review history of their article (what does this mean?). If published, this will include your full peer review and any attached files.

Reviewer #1: No

Reviewer #2: No

---

## [Author Response · Author response to Decision Letter 1]

25 Jan 2022

PONE-D-21-08432R1

Factors associated with female age at first marriage: an analysis using all waves of the Pakistan Demographic and Health Survey (revised version)

Response to editor and reviewers

The reviews of the revised version of the paper made only one comment. This concerns the decision not to include analysis of the women aged 15-24 years for the 2012-2013 and 2017-2018 Demographic and Health Surveys. The Reviewer disagreed with this decision. In our response with the first revision, we explained that one of our purposes is to analyse the trend in the age at first marriage in Pakistan over time. If we include the women aged 15-19 years in our analysis all of them will report ages at marriage below 20 years, and their average age at marriage will be well below 20 years. The women in the same birth cohort—that is, who would have been aged 15-19 years at the date of the survey but who will marry at ages above 20 years will not be included in the sample because they were not interviewed.

The Reviewer wrote: 

‘This argument is not convincing. The generalization of results in any survey-based study is restricted to the particular sample selected for the survey. Since the analysis is over time, the claimed inclusion of females who decide to marry above say age 20 is adequately presented in the subsequent survey. Even if this argument is accepted for the age bracket 15-19 [years], it is totally unacceptable for age 20-24. The study itself claims in l. 237 that “In general, more than 90% of the females marry before the age of 25 years”, so inclusion of this age bracket is very less likely to create any downward bias in the estimates.’

We disagree with the Reviewer on two counts: (1) about the generalisation of our results; and (2) about the downward bias in the estimates.

(1) The point made by the Reviewer is true in the sense that we cannot make inferences out of sample. However, we are not trying to do this. We are wanting to analyse the factors associated with the age at marriage among groups of women who have had every chance to get married. We also want to compare results from the 2017-2018 and 2012-2013 surveys with those in analyses of earlier surveys which only used data for women aged 25-49 years. In order to achieve comparability and to analyse only women who have had every chance to get married, we restrict attention to women aged 25-49 in the 2012-2013 and 2017-2018. We accept that this reduces our sample size, and it also means that we cannot make inferences or generalisations about women who were aged below 25 years in 2017-2018, but this is a price we are happy to pay. 

(2) We can demonstrate that the inclusion of women aged 15-19 years and 20-24 years results in a downward bias in our estimate of the trend. Our data come from four separate Demographic and Health Surveys, taken in 1990-1991, 2006-2007, 2012-2013 and 2017-2018. In each of these surveys we have data for a set of age cohorts and for each age cohort we can calculate the mean age at marriage. We can also calculate the average calendar year in which the women in each age cohort in each survey married. For example, the 2017-2018 survey took place over the period November 2017 – April 2018. On average, therefore, women were interviewed around the end of January 2018. The women aged 25-29 years in this survey were, on average, aged 27.5 years at the end of January 2018, and hence they were born, on average, at the end July 1990. They had an average age at marriage of 19.43 years, meaning that they married, on average, at the beginning of 2010. We can similarly calculate the average date of marriages for other age cohorts in other surveys. Plotting the average age at marriage of each age cohort against the average date of the relevant marriages allows us to track the trend over time in the mean age at marriage. The results are shown for all four surveys in the diagram below.

[It is not possible to present the diagram here: you are kindly refer to download/view the file named as 'Response to editor and Reviewer-second revision'] 

The black circles denote ages at marriage computed from data for women aged 25-49 years in the four surveys. The squares denote ages at marriage computed from data for women aged 20-24 and 15-19 years in the 2012-2013 and 2017-2018 surveys. It is clear that using data for woman aged 20-24 and 15-19 years will create a downward bias in our estimate of the trend in the age at marriage in recent years. The downward bias is, as the Reviewer correctly points out, less severe for women aged 20-24 years than for women aged 15-19 years, but it is nevertheless present for both age groups. 

We have not included a lengthy discussion of this issue in the second revision of our paper as the existence of these selection effects and the problems they can cause is a standard and well-known feature of the analysis of retrospective survey data (see, for example, the discussion in A. Hinde, Demographic Methods, London, Arnold, 1998, pp. 135-6). 

However, we have made the following changes to the paper to try to explain why we do not analyse data for women aged under 25 years at the time of the 2012-2013 and 2017-2018 surveys: 

• we have added a new Table 2 which includes estimates of the mean age at marriage for women aged 15-19 and 20-24 years in the 2012-2013 and 2017-2018 surveys

• we have renumbered the subsequent tables

• we have added a discussion of the results in the new Table 2 as follows:

‘Table 2 shows the mean age at first marriage for currently married women in the four surveys according to the age of the women at the time of the survey. The mean age at marriage for women in the 2017-18 survey was between 19.4 and 20.0 years for all age cohorts over 25 years. For younger age cohorts the mean age at marriage was substantially lower (18.16 years for those aged 20-24 years and only 16.12 years for those aged 15-19 years). These lower ages at marriage for the younger age cohorts are selection effects. Women aged 15-19 years who report an age at marriage in the Demographic and Health Surveys must all have married at ages less than 20 years; those women in the same cohort who are yet to marry will not report an age at marriage. As a consequence, the estimates of the mean age at marriage for this age cohort based on the data in the survey are underestimates of the mean age at marriage that would be reported by the women in the same age cohort were they to be interviewed when those who will eventually marry have all married. The same is true to a lesser extent for women aged 20-24 years. but not for older women, as the vast majority of women in Pakistan will marry before their 25th birthdays. Because of this, we exclude women aged under 25 years from the analysis which follows.’

• Because we now discuss the mean ages at marriage for women in different age cohorts at an earlier point in the paper, it made sense to us to move Figure 2 which examines the time trend so that it comes before Figure 1. We have accordingly done this, and renumbered these figures accordingly.

---

## [Decision Letter · Decision Letter 2]

14 Feb 2022

Factors associated with female age at first marriage: an analysis using all waves of the Pakistan Demographic and Health Survey

PONE-D-21-08432R2

Dear Dr. Nasir,

We’re pleased to inform you that your manuscript has been judged scientifically suitable for publication and will be formally accepted for publication once it meets all outstanding technical requirements.

Kind regards,

Faisal Abbas, PhD

Academic Editor

PLOS ONE

Additional Editor Comments (optional):

Accept.

Reviewers' comments:

Reviewer's Responses to Questions

**Comments to the Author**

1. If the authors have adequately addressed your comments raised in a previous round of review and you feel that this manuscript is now acceptable for publication, you may indicate that here to bypass the “Comments to the Author” section, enter your conflict of interest statement in the “Confidential to Editor” section, and submit your "Accept" recommendation.

Reviewer #1: All comments have been addressed

Reviewer #2: All comments have been addressed

2. Is the manuscript technically sound, and do the data support the conclusions?

Reviewer #1: Yes

Reviewer #2: Yes

3. Has the statistical analysis been performed appropriately and rigorously? 

Reviewer #1: Yes

Reviewer #2: Yes

4. Have the authors made all data underlying the findings in their manuscript fully available?

Reviewer #1: Yes

Reviewer #2: Yes

5. Is the manuscript presented in an intelligible fashion and written in standard English?

Reviewer #1: Yes

Reviewer #2: Yes

6. Review Comments to the Author

Reviewer #1: (No Response)

Reviewer #2: The author has addressed my comments in revised manuscript .Therefore, it may be accepted for publication.

7. PLOS authors have the option to publish the peer review history of their article (what does this mean?). If published, this will include your full peer review and any attached files.

Reviewer #1: No

Reviewer #2: No

---

## [Editor Report · Acceptance letter]

7 Mar 2022

PONE-D-21-08432R2 

Factors associated with female age at first marriage: an analysis using all waves of the Pakistan Demographic and Health Survey 

Dear Dr. Nasir:

I'm pleased to inform you that your manuscript has been deemed suitable for publication in PLOS ONE. Congratulations! Your manuscript is now with our production department. 

Kind regards, 

on behalf of

Dr. Faisal Abbas 

Academic Editor

PLOS ONE